# Transcriptome Analysis of the Development of Pedicel Abscission Zone in Tomato

**Xiufen Dong** [1,†]**, Yue Wang** [2,†]**, Yanyan Yan** [1] **and Huasen Wang** [2,*]

1   Collaborative Innovation Center for Efficient and Green Production of Agriculture in Mountainous Areas of Zhejiang Province, College of Horticulture Science, Zhejiang A&F University, Hangzhou 311300, China
2   Key Laboratory for Quality and Safety Control of Subtropical Fruits and Vegetables, Collaborative Innovation Center for Efficient and Green Production of Agriculture in Mountainous Areas of Zhejiang Province, Ministry of Agriculture and Rural Affairs, College of Horticulture Science, Zhejiang A&F University, Hangzhou 311300, China
*   Correspondence: wanghs@zafu.edu.cn
†   These authors contributed equally to this work.

**Abstract:** Plant organ abscission is a common phenomenon that occurs at a specific position called the abscission zone (AZ). The differentiation and development of the pedicel AZ play important roles in flower and fruit abscission, which are of great significance for abscission in tomatoes before harvest. Previous studies have reported some genes involved in AZ differentiation; however, the genes regulating pedicel AZ cell development in tomatoes after AZ differentiation remain poorly understood. In this study, transcriptome analyses of tomato pedicel AZ samples were performed at 0, 5, 15, and 30 days post-anthesis (DPA). Pedicel AZ growth was mainly observed before 15 DPA. A principal component analysis and a correlation analysis were carried out in order to compare the repeatability and reliability for different samples. We observed 38 up-regulated and 31 down-regulated genes that were significantly altered during 0 to 5 DPA, 5 to 15 DPA, and 0 to 15 DPA, which may play key roles in AZ cell enlargement. GO and KEGG enrichment analyses of the selected DEGs under all three different comparisons were conducted. Auxin-signaling-related genes were analyzed, as well as AUX/IAA, GH3, and small auxin up-regulated RNA (SAUR) gene expression patterns. The presented results provide information on pedicel AZ development, which might help in regulating flower or fruit pedicel abscission in tomato production facilities.

**Keywords:** tomato; pedicel AZ development; transcriptome analysis; auxin signaling

## 1. Introduction

The tomato (*Solanum lycopersicum* L.) is a common and popular vegetable due to its rich nutrients, the various ways to eat it, and its pleasing flavor. Flower and fruit abscission before harvest are two of the main factors affecting the tomato yield in production facilities. The location where a flower or fruit separates from the main plant is called the abscission zone (AZ). The AZ comprises layers of functionally specialized cells with morphologically distinct features, such as smaller, square-shaped cells that are interconnected by branched plasmodesmata and dense cytoplasm [1–4]. The development of a functional AZ is a prerequisite for plant organ abscission. The development of a pedicel AZ is of great significance for flower and fruit pedicel abscission in tomatoes.

The pedicel cells are initially found in the epidermal and cortical region. With the progression of a flower bud's development, the cell division gradually spreads to the vascular bundle region and, finally, to the central parenchyma region [5]. A number of genes were reported to be involved in the differentiation and development of AZs [6–13]. Among these genes, the transcription factor MADS-box plays an important role in the differentiation and development of plant AZs. One of the members of the MADS-box family, *jointless*, was first isolated as a functional gene involved in the development of plant AZs.

Since tomato plants with the *jointless* mutation fail to develop AZs on their pedicels, the abscission of flowers or fruit does not occur normally [6]. Another MADS-box, *MBP21*, has shown a similar function in AZ formation; the loss of function of *MBP21* led to a *jointless-2* phenotype in tomatoes [8,13]. *Macrocalyx* (*MC*) is also a member of the MADS-box family that has been confirmed to play a key role in pedicel AZ development in tomatoes [10]. In Arabidopsis, seedstick (STK) is required for the normal development of the funiculus—an umbilical-cord-like structure that connects the developing seed to the fruit—and for the dispersal of seeds when the fruit matures [7]. Furthermore, *blade-on-petiole* genes have been verified to play an essential role in abscission formation in Arabidopsis; the loss of function of *BOP1* and *BOP2* resulted in no differentiation of floral and leaf AZs, suggesting that BOP proteins are essential for the establishment of AZ cells [9,11,12].

Pedicel AZ development includes cell division and cell expansion. After the cells have differentiated into AZ cells, one of the main reasons for pedicel growth is cell expansion. The plant hormone auxin presents various functions in plant growth and development, regulating the fundamental cellular processes of expansion, division, and differentiation [14–16]. One of the most striking effects of auxin is rapidly mediating changes in cell expansion [17]. The acid growth theory supposes that auxin activates plasma membrane (PM) $H^+$ ATPases, resulting in proton efflux. The decreased pH of the cell wall matrix solution alters the activity of proteins that modify the cell wall, leading to changes in wall extensibility. Furthermore, elevated PM $H^+$ ATPase activity hyperpolarizes the PM and increases the energy required for solute uptake (which is necessary for the maintenance of water uptake and, therefore, the turgor pressure), forcing the expansion of the wall [18–20]. Auxin-regulated cell expansion has also been found to play a crucial role in leaf epinasty and storage organ expansion [21,22]. SAUR, which is a member of the auxin signaling pathway, has been reported to be involved in cell expansion through the regulation of PM $H^+$ ATPase activity [23,24]. Transmembrane kinase (TMK) auxin-signaling proteins interact with plasma membrane $H^+$ ATPases, inducing their phosphorylation, which is required for auxin-induced $H^+$-ATPase activation, apoplastic acidification, and cell expansion [25]. All of the above results indicate that auxin plays a key role in plant cell wall expansion. In addition to auxin, other hormones are also involved in cell expansion, cell elongation, and cell division. Brassinosteroids (BR) were reported to stimulate the pollen tube growth and hypocotyl elongation of pak choi (*Brassica rapa chinensis*) through cell and cell wall expansion [26–28]. Cytokinin regulated cell proliferation by influencing cell division and/or differentiation [29]. There was shown to be crosstalk between gibberellin (GA) and auxin in cell elongation; auxin promoted the degradation of Della in root cells in response to GA, which is a prerequisite for GA-induced root elongation [30].

For this study, we performed transcriptome analyses of pedicel AZs in tomatoes and aimed to determine the genes related to the development of the pedicel AZ. The results provide further information regarding the regulation of flower and fruit abscissions in tomato production.

## 2. Materials and Methods

### 2.1. Plant Materials and Sample Preparation

Tomato plants (Solanum lycopersicum L. cv. "micro-Tom") were grown in a greenhouse at 25/18 °C with a 16/8 h light cycle (day/night). To each flower, a small tag—which was labeled with the flower open date (i.e., the beginning of the anthesis stage)—was attached at 10:00 a.m. every day. Samples were collected at 0, 5, 15, and 30 days post-anthesis (DPA) from different plants that grew uniformly. The plants were grown at the same time and under the same conditions. At each stage, the pedicel abscission zones were cut into small segments (about 3 mm) using a sharp blade, and they were then immediately frozen in liquid nitrogen. The segments of pedicel AZs were weighed with a balance. Each sample weighted at least 1 g. Three replicates of each stage were considered.

### 2.2. AZ Diameter Measurement

The diameter of the AZ was measured using a vernier caliper at 10:30 a.m. each day. At least 30 AZs were tested at each time point. Each measurement was recorded in an Excel spreadsheet. The statistical significance was determined using one-way ANOVA and is indicated by asterisks (*) in the diagram.

### 2.3. mRNA Sequencing by Illumina HiSeq

The total RNA of each sample was extracted using TRIzol Reagent (Invitrogen), an RNeasy Mini Kit (Qiagen). The total RNA of each sample was quantified and checked for quality using an Agilent 2100 Bioanalyzer (Agilent Technologies, Palo Alto, CA, USA), NanoDrop (Thermo Fisher Scientific Inc, Waltham, MA, USA) and 1% agarose gel. For the libraries' construction, please refer to the Illumina sequencing method [31]. The sequences were processed on an Illumina HiSeq™ 2500 platform and analyzed by GENEWIZ (GENEWIZ, Inc. 115 Corporate Boulevard, South Plainfield, NJ, USA). The quality control was processed using Trimmomatic (v0.30). The sequencing error rate distribution, GC content distribution, and sequencing data filtering were done successively. In order to obtain the clean data, the adapter reads, bases containing N, and low-quality reads were sequentially removed. For reads mapping, the reference genome sequences and gene model annotation files of relative species were first downloaded from the genome website (https://solgenomics.net, accessed on 15 April 2019) using Current Tomato Genome version SL4.0 and Annotation ITAG4.0. Next, Hisat2 (v2.0.1) was used to index the reference genome sequence. Finally, the clean data were aligned to reference the genome via the software Hisat2 (v2.0.1) [32].

### 2.4. Differential Expression Analysis

For the differential expression analysis, we used the DESeq Bioconductor package—a model based on the negative binomial distribution. After an adjustment using the approach of Benjamini and Hochberg for controlling the false discovery rate, the P-values for the genes were determined [33,34].

### 2.5. Principal Component Analysis

A principal component analysis provides a way to visualize sample-to-sample distances. In this ordination method, the data points (here, the samples) are projected onto a 2D plane and spread out in the two directions that explain most of the variation in the data; the *x*-axis is the direction that separates the data points the greatest amount (the values of the samples in this direction are in terms of PC1), while the *y*-axis denotes the direction (which must be orthogonal to the first direction) that separates the data the second-greatest amount (the values of the samples in this direction are in terms of PC2). The percentage of the total variance that is contained in each direction is printed on the axis label. Note that these percentages generally do not add to 100%, as there are more dimensions that explain the remaining variance (although each of these remaining dimensions will explain fewer than the two that are presented). This analysis was performed using the OmicShare tool, a free online platform for data analysis (https://www.omicshare.com/tools, accessed on 8 June 2022).

### 2.6. Correlation Analysis

The correlation of each of the two samples was calculated using the Pearson method. In this method, the correlation values range between −1 and 1. If the correlation value is between −1 and 0, the two samples have a negative correlation. On the contrary, if the correlation value is between 0 and 1, the correlation between the two samples is positive. If the correlation value is 0, there is no correlation between the two samples. The closer that the absolute value of the correlation coefficient is to 1, the stronger that the correlation is between the two variables; meanwhile, the closer that the correlation coefficient is to zero,

the weaker the correlation is between the two variates. This analysis was also performed using the OmicShare tool (https://www.omicshare.com/tools, accessed on 10 June 2022).

### 2.7. GO and KEGG Enrichment Analysis

GO Term Finder was used to identify gene ontology (GO) terms in order to annotate a list of enriched genes with significant P-values (i.e., less than 0.05). KEGG (Kyoto Encyclopedia of Genes and Genomes) is a collection of databases dealing with genomes, biological pathways, diseases, drugs, and chemical substances (http://en.wikipedia.org/wiki/KEGG, accessed on 13 June 2022). We used in-house scripts to enrich significantly differentially expressed genes in KEGG pathways [31].

### 2.8. Quantitative RT-PCR

Tomato pedicel AZs at 0 DPA, 5 DPA, 15 DPA and 30 DPA were collected for total RNA isolation. The total RNAs were extracted using the TRIzol reagent, following the protocol provided by the manufacturer (Invitrogen), and treated with DNase I (Thermo Scientific). The total RNAs from each sample were used for first-strand cDNA synthesis in a final volume of 20 μL. For real-time quantitative PCR (qRT-PCR), a method with two steps of Takara real-time PCR was used to amplify the representative auxin-related genes. Each experiment was performed independently three times (technical replicates) with three biological samples. Each biological sample's Ct value was the average of the three technical replicates. Each sample's Ct value was the average of the three biological Ct values. The data analysis for qRT-PCR was performed following the ΔΔCt model [35]. The relative expression of qRT-PCR was calculated according to the formula $2^{[Ct(actin) - Ct(gene)]}$. The error bar was the standard deviation of the three biological replicates. The primers used in this study are listed in Supplemental File Table S7. The raw Ct values are provided in Supplemental File Table S8.

## 3. Results

### 3.1. Development of Pedicel AZ in Tomato

The different development stages of tomato pedicels are shown in Figure 1a. The pedicel AZ size was measured at 0 (the day of flower opening), 5, 15, and 30 days post-anthesis (DPA). About 30 pedicels of tomato plants were measured using Vernier calipers. The diagram shows that the pedicel AZs significantly changed between the day of flower opening and 15 DPA (Figure 1b). The diameter of the AZ was about 1.2, 2.1, 3.2, and 3.3 mm at 0, 5, 15, and 30 DPA, respectively. There were significant differences between 0 and 5 DPA and 5 and 15 DPA, but there was no significant difference between 15 and 30 DPA. The results indicate that the main period of pedicel AZ growth or size expansion is from 0 to 15 DPA, and the AZ diameter almost reached its peak at 15 DPA.

### 3.2. Transcriptome Analysis of Pedicel AZ at Different Development Stages in Tomatoes

The tomato pedicel AZs of 0, 5, 15, and 30 DPA were collected for transcriptome sequencing. The principal component analysis showed that the distance between the three replicates of each group was relatively small, which meant that the interclass replicates were reliable (Figure 1c). The samples at 0, 5, and 15 DPA showed large distances between the groups; however, the samples at 15 and 30 DPA were very close to each other (Figure 1c). This result indicates that the pedicel AZs at 15 DPA were similar to those at 30 DPA. A correlation analysis of the 12 samples also showed a similar result (Figure 1d). The correlation coefficients of the interclass replicates were all very close to one, which meant that strong correlations existed in the two samples. The correlation coefficients between the samples at 15 and 30 DPA were also very close to one (Figure 1d), confirming the PCA results and being consistent with pedicle AZ development in tomatoes. According to the process of pedicel AZ growth, we separated the main stage containing the three time points of 0, 5, and 15 DPA into three parts: 0–5 DPA, 5–15 DPA, and 0–15 DPA. For 0–5 DPA, a total of 4204 DEGs, including 2277 up- and 1927 down-regulated genes, were identified

(Figure 1e). From 5–15 DPA, 5526 DEGs, including 2436 up- and 3090 down-regulated genes, were identified (Figure 1e). From 0–15 DPA, 4591 DEGs, including 2132 up- and 2459 down-regulated genes, were identified (Figure 1e). The DEGs of different comparisons were showed in supplementary files Tables S1–S6.

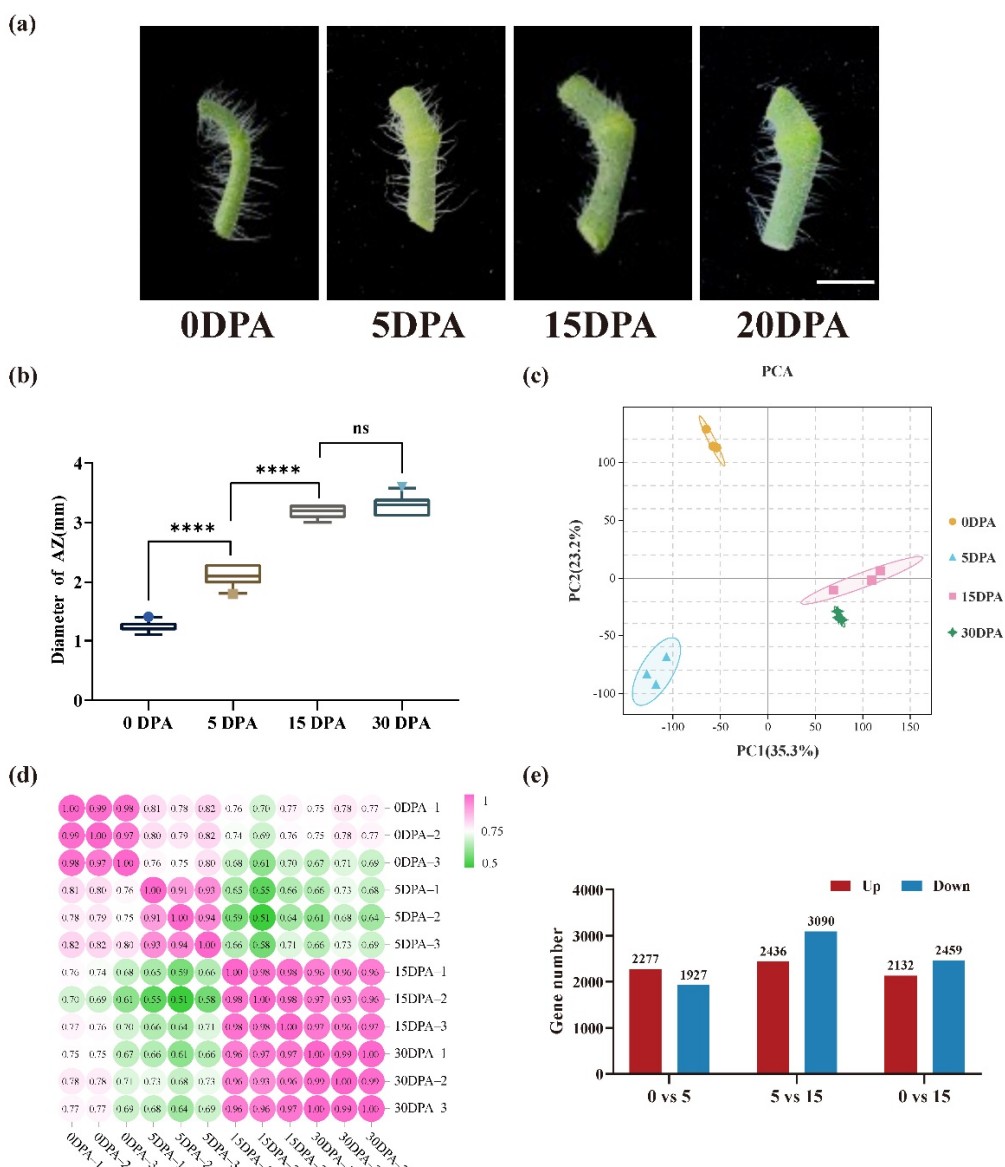

**Figure 1.** Tomato pedicels, AZ size, and transcriptome analysis between different development stages. DPA, days post-anthesis. (**a**) The development of tomato pedicels. (**b**) AZ diameter in different developmental stages: ****, extremely significant (*p* < 0.0001); ns, no significance. (**c**) Principal component analysis of the transcriptome samples. (**d**) Correlation analysis between the twelve samples. (**e**) Number of differentially expressed genes in three different comparisons.

### 3.3. KEGG Enrichment Analysis of the DEGs

The DEGs in the three different developmental stages were then classified into various KEGG signaling and metabolic pathways (Figure 2). From 0–5 DPA, the DEGs were significantly enriched in 19 pathways; from 5–15 DPA, they were enriched in 20 pathways; and from 0–15 DPA, they were enriched in 23 pathways. Five KEGG pathways—"pentose and glucuronate interconversions", "ascorbate and aldarate metabolism", "cysteine and methionine metabolism", "phenylalanine metabolism", and "cyanoamino acid metabolism"—were significantly changed during 0–5 DPA; six KEGG pathways—"fatty acid biosynthesis",

"glutathione metabolism", "biotin metabolism", "DNA replication", "homologous recombination", and "AGE-RAGE signaling pathway in diabetic complications"—were significantly changed during 5–15 DPA; seven pathways—"photosynthesis", "alpha-linolenic acid metabolism", "glycosphingolipid biosynthesis-lacto and neolacto series", "glycosphingolipid biosynthesis-globo and isoglobo series", "glyoxylate and dicarboxylate metabolism", "brassinosteroid biosynthesis", and "carotenoid biosynthesis"—were not significantly changed during both 0–5 DPA and 5–15 DPA, whereas they were significantly altered during 0–15 DPA; and two pathways—"amino sugar and nucleotide sugar metabolism" and "plant-pathogen interaction"—were changed during both 0–5 DPA and 5–15 DPA, but not during 0–15 DPA. Finally, eight pathways—"starch and sucrose metabolism", "linoleic acid metabolism", "phenylpropanoid biosynthesis", "flavonoid biosynthesis", "stilbenoid, diarylheptanoid, and gingerol biosynthesis", "biosynthesis of secondary metabolites", "MAPK signaling pathway-plant", and "plant hormone signal transduction"—were all significantly changed in all three of the developmental stages.

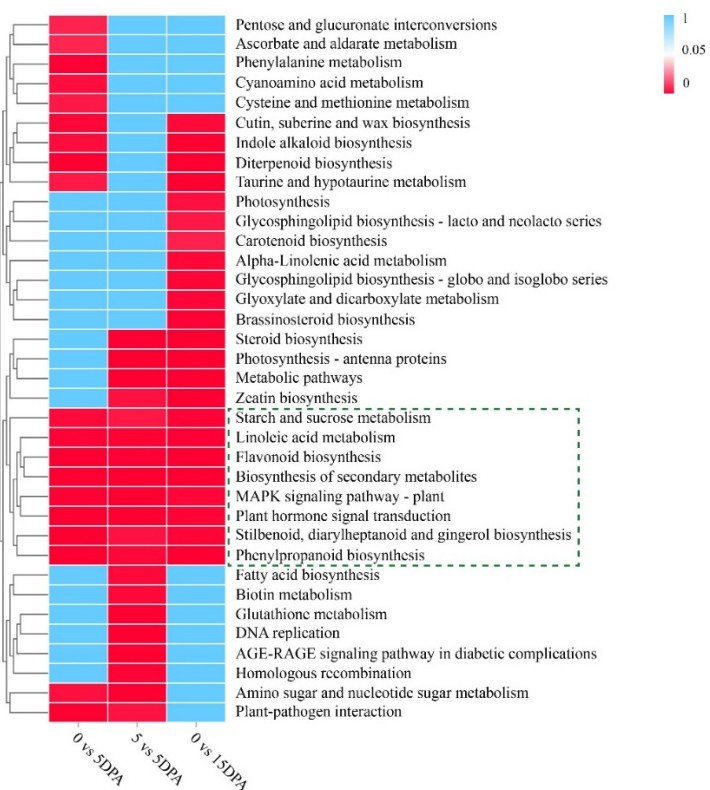

**Figure 2.** KEGG enrichment analysis of DEGs under three different comparisons. The significant *p*-values of each KEGG term under different comparisons are shown with a heat map. The green box indicates that the metabolic pathway was significantly changed between different developmental stages.

### 3.4. Differentially Expressed Genes That Were Selected under Three Comparisons

With the main growth stage separated into three parts, as above, we wished to determine the DEGs that had all changed under the three comparisons. For the up-regulated genes, 125 DEGs were changed under all three comparisons, while 1504, 1339, and 389 DEGs were only changed for 0–5 DPA, 5–15 DPA, and 0–15 DPA, respectively (Figure 3a). For the down-regulated genes, 167 DEGs were changed under all three comparisons, while 1103, 1710, and 422 DEGs were only changed for 0–5 DPA, 5–15 DPA, and 0–15 DPA, respectively (Figure 3b). Genes were discarded if their absolute values of fold change were less than five. For the up-regulated genes, if a gene's fpkm value was less than 1 at 15 DPA, it was abandoned; however, for the down-regulated genes, if a gene's fpkm value was less than 1 at 0 DPA, it was also abandoned, as it may barely play a role during the development of the

pedicel AZ in tomatoes. Based on these criteria, we selected 38 up-regulated (Table 1) and 31 down-regulated (Table 2) genes. The ID, fpkm values, and descriptions of these genes are provided in Tables 1 and 2.

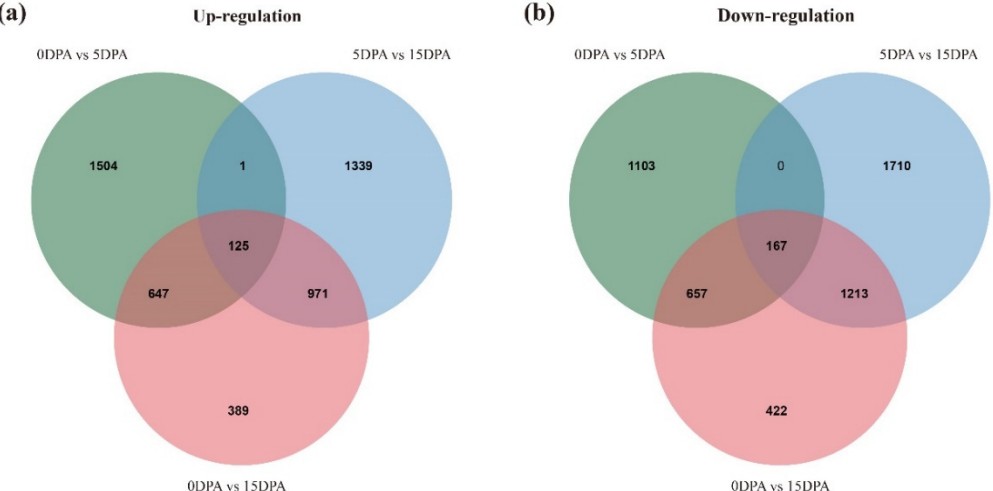

**Figure 3.** Venn diagram showing the numbers of up- and down-regulated genes between the three different comparisons. (**a**) Venn diagram showing the numbers of up-regulated genes between the three different comparisons. (**b**) Venn diagram showing the numbers of down-regulated genes between the three different comparisons.

**Table 1.** The 38 significantly up-regulated genes under all three comparisons.

| Gene ID | Fpkm-F | Fpkm-5 | Fpkm-15 | Significant | Description |
|---|---|---|---|---|---|
| Solyc12g006750 | 0 | 3.10 | 173.10 | Up | Glutathione S-transferase zeta 1 |
| Solyc04g080540 | 0.44 | 27.95 | 129.26 | Up | DNA polymerase epsilon catalytic subunit A, putative |
| Solyc08g079270 | 0.77 | 45.09 | 130.87 | Up | R2R3MYB transcription factor 42 |
| Solyc09g074890 | 1.61 | 72.17 | 239.28 | Up | Rapid alkalinization factor |
| Solyc02g090120 | 2.89 | 117.58 | 240.35 | Up | Low quality: Inositol 1,4,5-trisphosphate receptor-interacting protein-like 2 |
| Solyc07g008240 | 0.62 | 16.34 | 41.20 | Up | Non-symbiotic hemoglobin 1 |
| Solyc11g012690 | 1.03 | 22.45 | 57.97 | Up | Heavy metal transport/detoxification superfamily protein |
| Solyc10g005010 | 1.03 | 20.42 | 49.83 | Up | NAC domain-containing protein, putative |
| Solyc07g055490 | 1.39 | 24.04 | 76.64 | Up | Cytochrome P450 |
| Solyc06g072550 | 0.59 | 9.60 | 109.96 | Up | CASP-like protein |
| Solyc07g042460 | 0.32 | 5.22 | 109.96 | Up | Respiratory burst oxidase, putative |
| Solyc11g069900 | 3.00 | 48.67 | 158.53 | Up | Alpha/beta hydrolase superfamily protein |
| Solyc02g094230 | 1.09 | 17.46 | 58.70 | Up | Low quality: Dynein heavy chain 1, axonemal |
| Solyc11g012705 | 0.68 | 10.00 | 28.37 | Up | Pollen Ole e 1 allergen and extensin family protein |
| Solyc07g053840 | 3.94 | 56.85 | 183.40 | Up | Leucine-rich repeat receptor-like protein kinase family |
| Solyc05g056170 | 8.12 | 109.98 | 223.71 | Up | Phenylalanine ammonia-lyase 2 |
| Solyc02g093270 | 8.59 | 115.79 | 313.50 | Up | Caffeoyl-CoA O-methyltransferase |
| Solyc04g076220 | 0.20 | 2.40 | 14.38 | Up | AT hook motif DNA-binding family protein |
| Solyc09g065890 | 0.30 | 3.43 | 7.79 | Up | Alpha/beta hydrolase superfamily protein |
| Solyc01g091700 | 0.13 | 1.44 | 4.22 | Up | F-box family protein |
| Solyc04g071085 | 3.09 | 33.10 | 351.47 | Up | Hydroxyproline-rich glycoprotein |
| Solyc12g042460 | 0.39 | 4.15 | 15.96 | Up | 4-coumarate—CoA ligase-like |
| Solyc01g111880 | 0.86 | 8.16 | 19.77 | Up | MAP kinase kinase kinase 11 |
| Solyc09g015350 | 0.35 | 3.19 | 8.37 | Up | Trichome birefringence-like 34 |
| Solyc12g005820 | 0.97 | 7.68 | 21.12 | Up | Basic helix-loop-helix (bHLH) DNA-binding superfamily protein |
| Solyc11g072920 | 0.14 | 1.09 | 2.26 | Up | Peroxidase |
| Solyc07g065330 | 1.74 | 12.82 | 26.00 | Up | Germin-like protein |
| Solyc01g099620 | 0.24 | 1.68 | 5.43 | Up | Respiratory burst oxidase, putative |
| Solyc06g084130 | 0.70 | 4.52 | 9.43 | Up | Bax inhibitor |
| Solyc02g084570 | 20.62 | 131.85 | 303.65 | Up | Cytochrome P450 family protein |
| Solyc07g054780 | 22.19 | 123.00 | 440.86 | Up | Low quality: Wound-responsive family protein |
| Solyc12g006560 | 0.88 | 4.83 | 51.75 | Up | Early nodulin-93 |
| Solyc12g088800 | 0.34 | 1.88 | 5.92 | Up | Lipase |
| Solyc10g084070 | 13.35 | 71.89 | 181.87 | Up | Low quality: Endoglucanase |

**Table 1.** *Cont.*

| Gene ID | Fpkm-F | Fpkm-5 | Fpkm-15 | Significant | Description |
|---|---|---|---|---|---|
| Solyc12g044950 | 12.54 | 67.54 | 851.34 | Up | Lipid desaturase |
| Solyc08g081690 | 1.37 | 7.19 | 25.45 | Up | NADPH oxidase |
| Solyc06g075660 | 1.79 | 9.13 | 19.08 | Up | MYB family protein |
| Solyc12g099190 | 15.84 | 79.65 | 173.51 | Up | Low quality: Invertase inhibitor |

Note: F, flower-opening stage (0 DPA); 5, 5 DPA; 15, 15 DPA.

**Table 2.** The 31 significantly down-regulated genes under all three comparisons.

| Gene ID | Fpkm-F | Fpkm-5 | Fpkm-15 | Significant | Description |
|---|---|---|---|---|---|
| Solyc05g015880 | 87.66 | 6.25 | 0.63 | Down | Regulator of nonsense transcript protein |
| Solyc09g091550 | 98.55 | 7.78 | 1.99 | Down | S-adenosyl-L-methionine: salicylic acid carboxyl methyltransferase |
| Solyc03g034390 | 1647.24 | 141.12 | 13.80 | Down | Lipid transfer protein |
| Solyc09g006010 | 218.07 | 19.50 | 2.62 | Down | Pathogenesis-related protein 1 |
| Solyc06g007370 | 71.51 | 6.80 | 2.54 | Down | Low quality: 6,7-dimethyl-8-ribityllumazine synthase |
| Solyc06g051680 | 33.61 | 3.20 | 0 | Down | Protein early flowering 4 |
| Solyc01g099150 | 33.94 | 3.35 | 0.65 | Down | Desiccation-related protein PCC13-62 |
| Solyc09g072750 | 505.81 | 50.45 | 12.27 | Down | Low quality: Carbohydrate-binding X8 domain superfamily protein |
| Solyc03g114560 | 65.92 | 7.058 | 0.28 | Down | Strictosidine synthase-like protein, putative |
| Solyc09g092715 | 186.43 | 20.23 | 3.48 | Down | RGG repeats nuclear RNA-binding protein B |
| Solyc03g005910 | 46.05 | 5.19 | 0.52 | Down | Lipase, GDSL |
| Solyc01g088300 | 117.22 | 13.72 | 1.38 | Down | Germin-like protein 1 |
| Solyc03g020040 | 718.69 | 84.41 | 21.22 | Down | Pin-II type proteinase inhibitor 69 |
| Solyc09g005260 | 63.83 | 7.70 | 1.82 | Down | Vacuolar cation/proton exchanger 3 |
| Solyc00g071180 | 21.55 | 2.77 | 0 | Down | Multicystatin |
| Solyc12g039070 | 73.40 | 10.58 | 0.87 | Down | Strictosidine synthase |
| Solyc07g056670 | 190.42 | 28.74 | 10.91 | Down | Gibberellin 2-oxidase 2 |
| Solyc02g078850 | 60.27 | 9.39 | 1.20 | Down | Glycine-rich protein |
| Solyc07g007240 | 202.25 | 32.30 | 7.11 | Down | Metallocarboxypeptidase inhibitor |
| Solyc03g113560 | 32.06 | 5.24 | 0.21 | Down | Basic helix-loop-helix (bHLH) DNA-binding superfamily protein |
| Solyc09g092760 | 223.22 | 36.78 | 6.60 | Down | RNA-binding (RRM/RBD/RNP motifs) family protein |
| Solyc09g014940 | 39.74 | 7.34 | 1.38 | Down | Wound-induced protein 1 |
| Solyc08g082210 | 68.83 | 12.83 | 2.75 | Down | AP2/EREBP transcription factor |
| Solyc03g113250 | 123.55 | 23.47 | 4.69 | Down | Nitrate transporter 1:2 |
| Solyc03g097820 | 27.69 | 5.27 | 1.82 | Down | bHLH transcription factor 022 |
| Solyc12g005400 | 36.63 | 7.07 | 2.64 | Down | Cyclic nucleotide-gated ion channel, putative |
| Solyc08g067030 | 2070.64 | 400.99 | 21.90 | Down | Transmembrane protein, putative (protein of unknown function, DUF642) |
| Solyc12g055810 | 10.89 | 2.13 | 0.46 | Down | P-loop containing nucleoside triphosphate hydrolase superfamily protein |
| Solyc10g009150 | 209.96 | 42.47 | 0 | Down | Organ-specific protein S2 |
| Solyc05g009270 | 15.68 | 3.22 | 0.96 | Down | 3-ketoacyl-CoA synthase |
| Solyc10g074440 | 138.10 | 29.38 | 4.57 | Down | Chitinase |

Note: F, flower-opening stage (0 DPA); 5, 5 DPA; 15, 15 DPA.

### 3.5. GO and KEGG Enrichment Analysis of the Key DEGs under Three Comparisons

The biological functions of the selected DEGs were annotated with GO terms under all three different developmental stages. For the up-regulated genes, 14 and 7 GO terms were in the categories associated with molecular functions and biological processes, respectively. The top three enriched GO terms in the molecular function category were "oxidoreductase activity", "DNA binding", and "protein binding", while those in the biological process category were "oxidation-reduction process", "lipid metabolic process", and "metabolic process" (Figure 4a). For the down-regulated genes, 3, 16, and 8 GO terms were part of the cellular component, molecular function, and biological process categories, respectively. The top three enriched GO terms in the cellular component category were "membrane", "extracellular region", and "integral component of membrane"; those in the molecular function category were "protein dimerization activity", "oxidoreductase activity", and "strictosidine synthase activity"; and those in the biological process category were "oxidation-reduction process", "transmembrane transport", and "biosynthetic process" (Figure 4b). The information of the DEGs that were annotated in each of the GO terms was showed in Table S9. The KEGG enrichment analysis showed that the DEGs under all three comparisons were enriched in 14 KEGG pathways (Figure 4c). The up-regulated genes were involved in nine KEGG pathways, where the top three pathways were "phenylpropanoid biosynthe-

sis", "metabolic pathways", and "biosynthesis of secondary metabolites". Meanwhile, the down-regulated genes were involved in five KEGG pathways, including "amino sugar and nucleotide sugar metabolism", "diterpenoid biosynthesis", "fatty acid elongation", "metabolic pathways", and "biosynthesis of secondary metabolites" (Figure 4c).

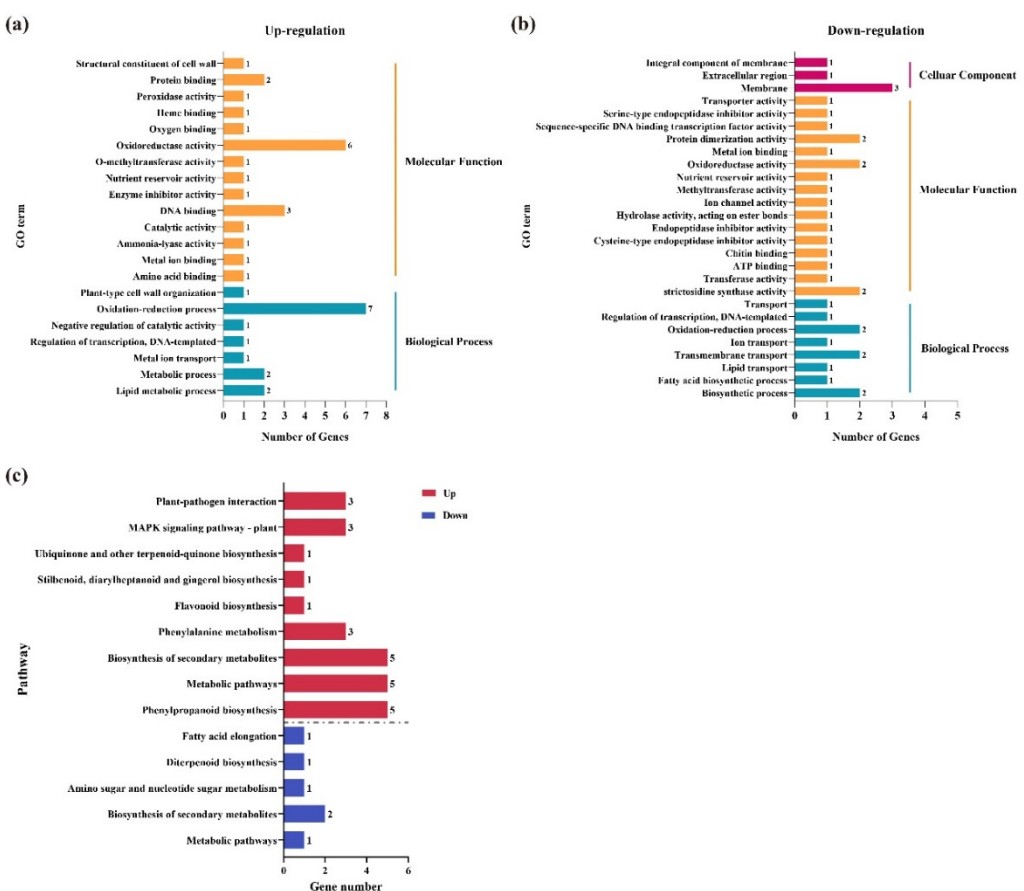

**Figure 4.** GO and KEGG enrichment analysis of the DEGs under all three different comparisons. The numbers of significantly up-regulated (**a**) and down-regulated (**b**) genes annotated with different GO terms under all three developmental stages are shown. (**c**) Classification of enriched KEGG pathways under the three developmental stages.

*3.6. Analysis of DEGs Involved in the Auxin Signaling Pathway*

Plant hormones are well-known to be involved in plant growth development. An enrichment analysis for all hormone-related genes involved in pedicel AZ development is shown in Figure S1. Among the hormone pathways, the DEGs were significantly enriched in the auxin signaling pathway. Auxin is one of the main factors affecting cell expansion. The auxin signaling pathway is depicted in Figure 5a. A number of auxin-related genes were identified in the three different comparisons. The enrichment analysis of the DEGs belonging to different classes of auxin response genes was showed in Figure S2. The numbers of DEGs belonging to AUX/IAA (K14484) and SAUR (K14488) were higher than those related to the other three families (Figures 5b and S1). In the AUX1 (13946) family, there were no up-regulated genes in any of the three stages. In the ARF (14486) family, only two genes were up-regulated during 0–5 DPA, while one gene was down-regulated during 5–15 DPA and 0–15 DPA. In the GH3 (14487) family, three and four genes were up-regulated during 0–5 DPA and 0–15 DPA, respectively, while one gene was down-regulated during 5–15 DPA (Figure 5b).

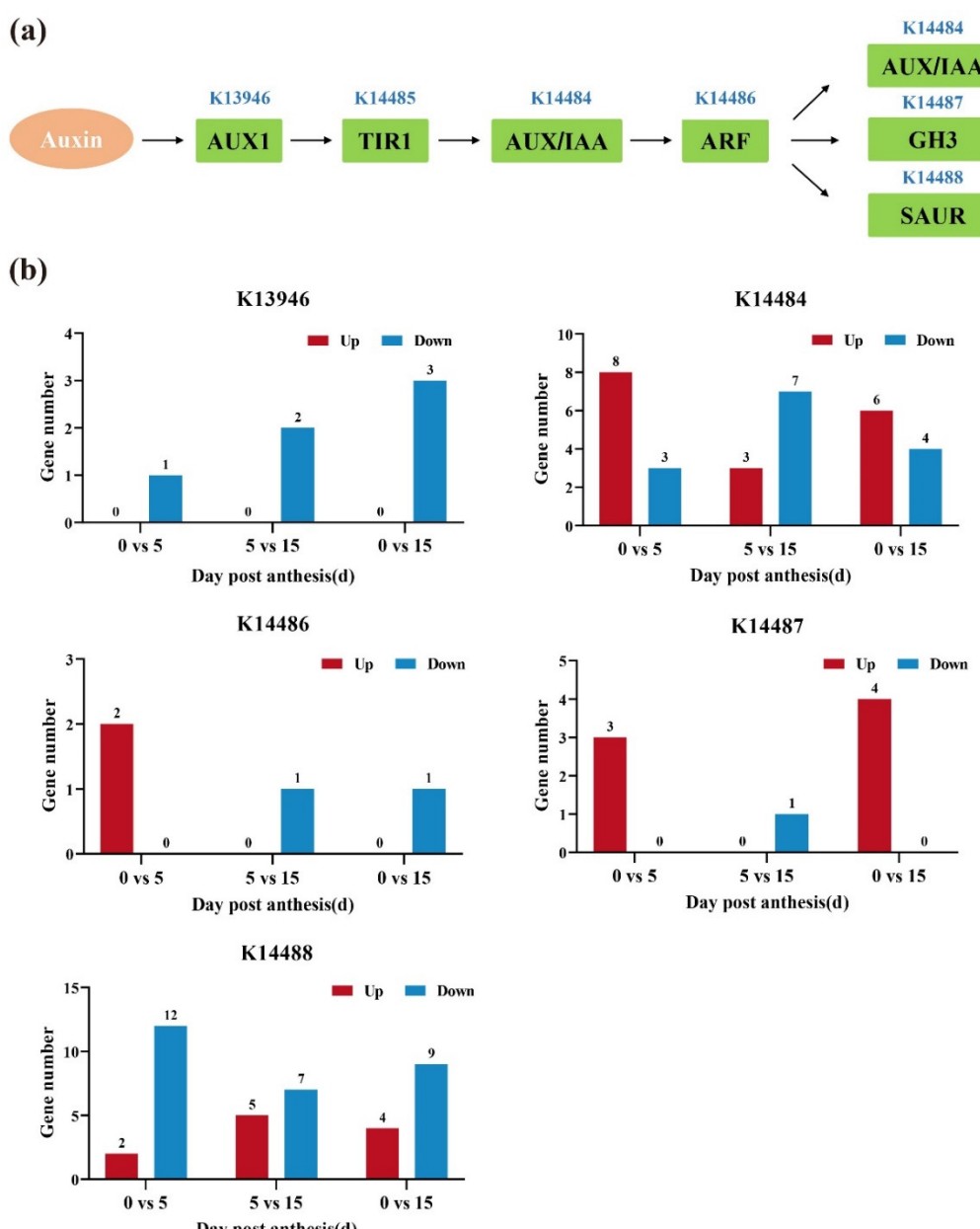

**Figure 5.** The transcript abundance of DEGs involved in the auxin signaling pathway in the three considered stages. (**a**) The auxin signaling pathway in tomatoes. (**b**) Numbers of DEGs involved in the auxin signaling pathway in all three stages.

*3.7. Expression Pattern of Auxin-Related Genes during All Three Development Stages*

The number of auxin-related genes significantly differed in the three different stages, and the expression patterns of these genes are shown in Figure 6. In the four AUX/IAA family members, the expression of three genes—Solyc12g096980 (IAA11), Solyc06g066020 (IAA36), and Solyc03g120380 (IAA19)—showed an increasing and then decreasing trend, with a peak at 5 DPA. The other gene, Solyc08g021820 (IAA29), also showed an increasing and then decreasing trend, but its highest expression was at 15 DPA (Figure 6a). The only GH3 family member, Solyc02g092820, also showed an increasing and then decreasing trend, with an expression pattern very similar to that of IAA11 during the pedicle AZ development in tomatoes (Figure 6b). In the SAUR family, the expression of two genes—Solyc08g079150 and Solyc01g110580—showed a similar trend, with both showing their highest expression at 0 DPA. The other gene, Solyc06g053260, showed a trend opposite to these two genes,

with a peak at 30 DPA (Figure 6c). The expression pattern of these genes were verified by qRT-PCR (Figure 7). All of these genes showed a similar expression pattern with the RNA-Seq data (Figure 6). Two AUX/IAA genes—Solyc12g096980 (IAA11) and Solyc06g066020 (IAA36) (Figure 7a)—and the GH3 member—Solyc02g092820 (Figure 7b)—showed an increasing and then decreasing trend, but the peak of these genes' expression were at 15DPA instead of the 5DPA shown by the RNA-Seq data.

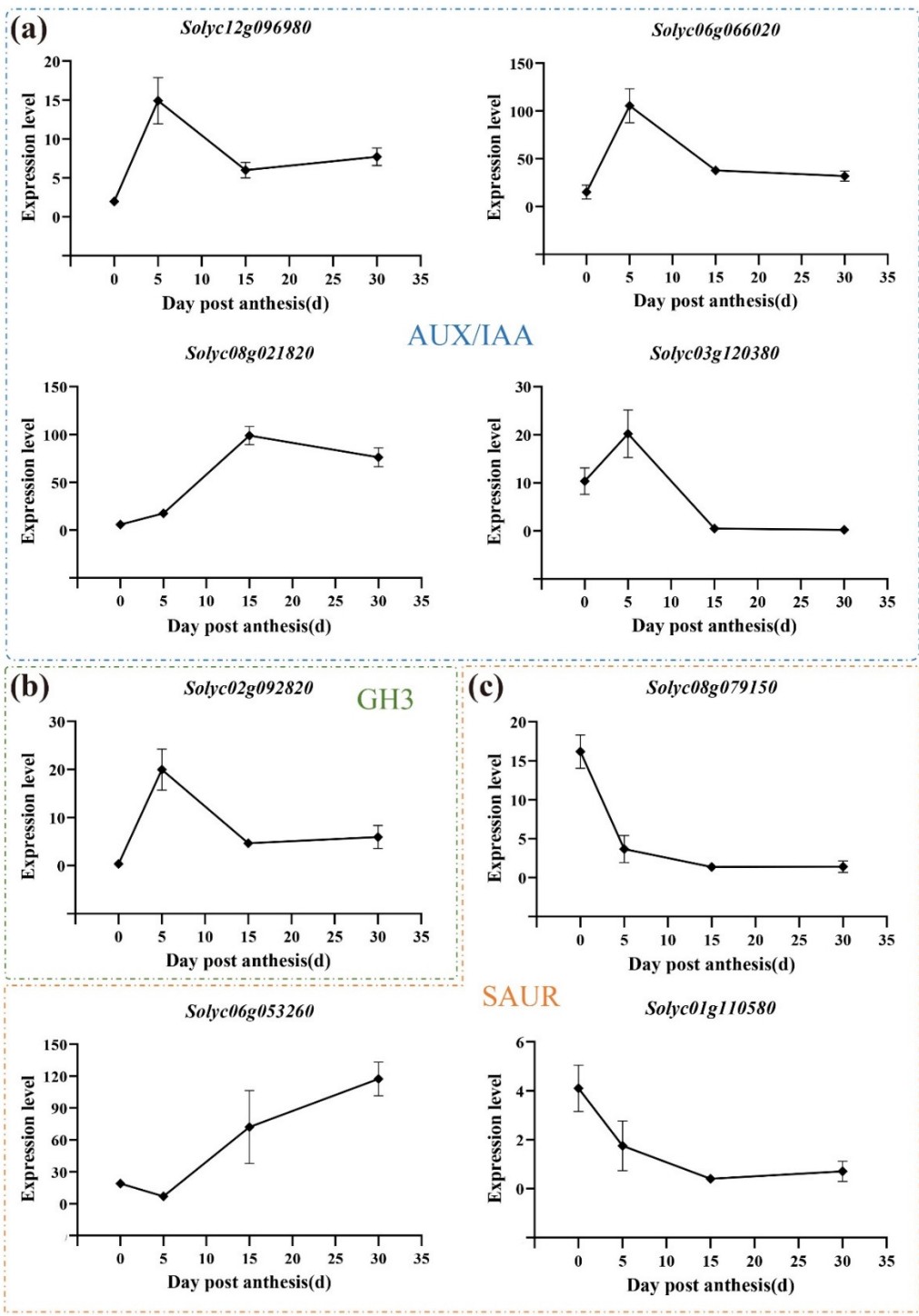

**Figure 6.** The expression levels of auxin-related genes at different developmental stages. (**a**) The blue box indicates the genes belonging to the AUX/IAA family. (**b**) The green box indicates the genes belonging to the GH3 family. (**c**) The orange box indicates the genes belonging to the SAUR family.

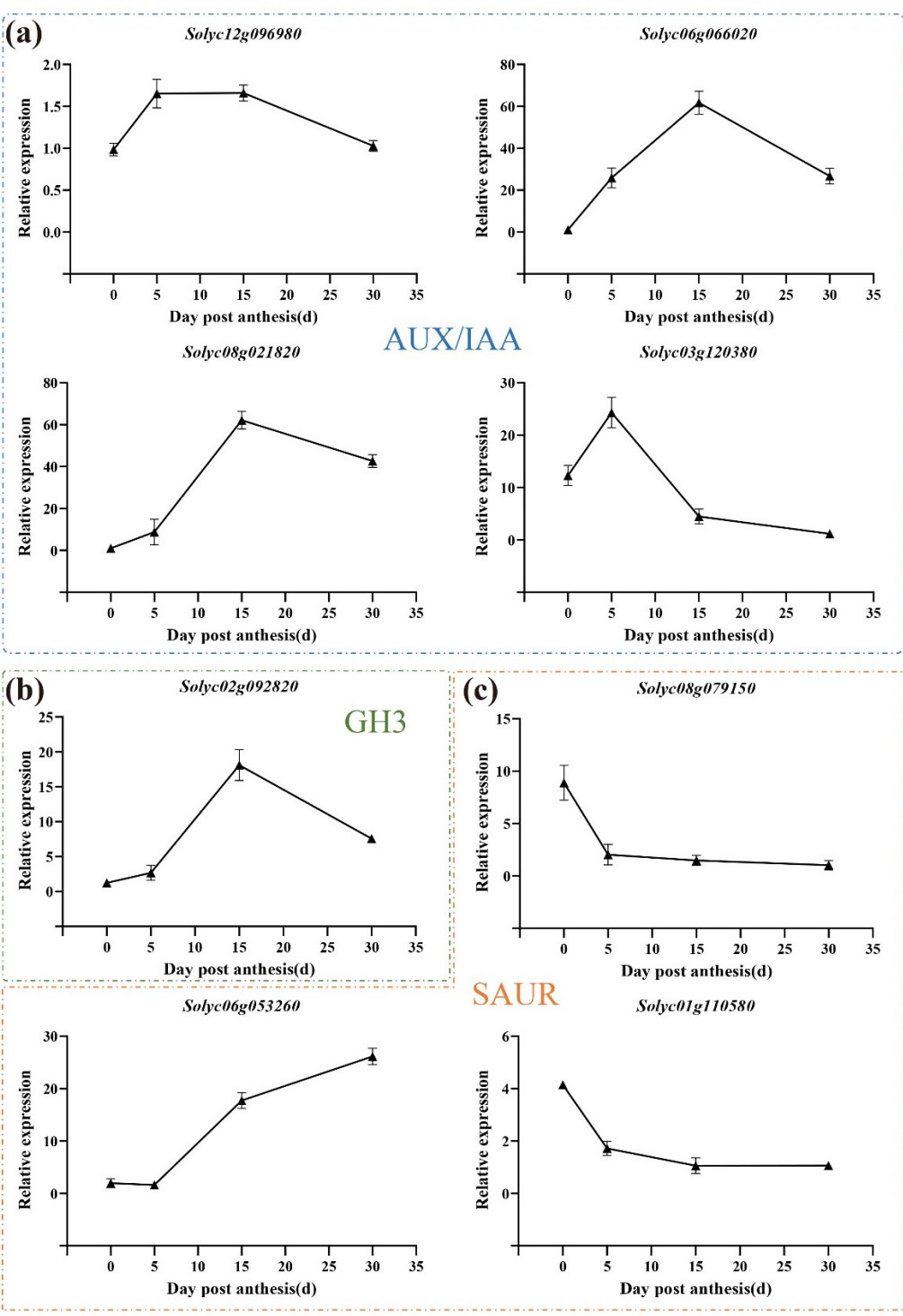

**Figure 7.** The relative expression of auxin-related genes at different developmental stages. (**a**) The blue box indicates the genes belonging to the AUX/IAA family. (**b**) The green box indicates the genes belonging to the GH3 family. (**c**) The orange box indicates the genes belonging to the SAUR family.

## 4. Discussion

Plant cells are surrounded by cell walls, which provide structural integrity but also spatially constrain cells and, therefore, must be modified to allow for cellular expansion [16], which is triggered by a high intracellular turgor pressure. The wall properties regulate the differential growth of the cell, resulting in a diversity of cell sizes and shapes. The acid growth theory supposes that a decrease in the pH of the cell wall matrix's solution alters

the wall's extensibility [18–20]. However, different cells have displayed distinct abilities to perceive acid; mature cells are less sensitive to acidic pH and extend less than young cells [18,36]. In this study, the enlargement of pedicel AZ size was mainly observed from 0 to 15 DPA (Figure 1a), and the growth of the pedicel AZ was slow after 15 DPA, indicating that the cells of the pedicel AZ at 15 DPA might be almost mature. The cells of the pedicel AZ after 15 DPA were less sensitive to acidic pH conditions, which led to cell expansion or to an organ's growth slowing or stopping. A correlation analysis of samples at different developmental stages confirmed this result: the coefficients of the correlation between the samples at 15 and 30 DPA were very close to one, which meant the samples at these two stages were very similar to each other (Figure 1c). Thus, our results indirectly verify the supposition that mature cells are less sensitive to acidic pH than young cells are.

It was interesting to find that seven pathways—"photosynthesis", "alpha-linolenic", "glycosphingolipid biosynthesis-lacto and neolacto series", "glyco-sphingolipid biosynthesis-globo and isoglobo series", "glyoxylate and dicarboxylate metabolism", "brassinosteroid biosynthesis", and "carotenoid biosynthesis"—were not significantly changed during both 0–5 DPA and 5–15 DPA, whereas they were significantly altered during 0–15 DPA (Figure 2). This suggested that the genes involved in these pathways might show the same trend between 0 and 5 DPA and between 5 and 15 DPA, such as when they were on the rise. The pathway increased both a little bit from 0 to 5 DPA and 5 to 15 DPA, but these two increases resulted in a significant rise overall from 0 to 15 DPA. Conversely, for the downward trend, the pathways might show similar results. Two pathways— "amino sugar and nucleotide sugar metabolism" and "plant-pathogen interaction"—were changed during both 0–5 DPA and 5–15 DPA, but not in 0–15 DPA (Figure 2). This result indicated that the two pathways might show a reverse trend between 0 and 5 DPA and between 5 and 15 DPA. This meant that when the pathway first rose and then fall or when it first fell and then rose, the high peak or the low peak was at 5 DPA but showed similar levels at 0 and 15 DPA. Based on this finding, the two pathways might play important roles on the early stage of pedicel AZ development in tomatoes.

Auxin regulates numerous processes in plant growth and development through auxin signal transduction [37,38]. This mechanism includes three core components: the F-box proteins TIR1/AFB1–AFB5, Aux/IAA transcriptional repressors, and activator class of auxin response factor (ARF) transcription factors [38–40]. In Arabidopsis, the AUX/IAA mutants *axr2/iaa7*, *axr5/iaa1*, *axr3/iaa17,* and *shy2/iaa3* presented cell expansion defects, indicating that auxin induces cell expansion through the degradation of AUX/IAAs [14,41]. In our study, eight, three, and six AUX/IAA genes were significantly up-regulated during 0–5 DPA, 5–15 DPA, and 0–15 DPA, respectively. The four AUX/IAA genes all showed increasing trends before 15 DPA, indicating that these four genes might play key roles in pedicel AZ development in tomatoes.

In addition to these three core components, the genes belonging to the AUX1, GH3, and SAUR families are also involved in the auxin signaling pathway (Figure 5a). Auxin is known to induce acid growth, which is regulated by the auxin-inducible SAUR proteins [24]. In this study, the number of DEGs belonging to the SAUR family involved in the auxin signaling pathway was the largest (Figure 5b). This suggests that SAUR genes might play an important role during pedicel cell development, which is consistent with the results of previous studies.

In conclusion, we analyzed gene expressions at the transcriptional level and selected some differentially expressed genes that may play an important role in pedicel AZ development. This study provides a foundation for research into plant organ abscission, as well as providing a reference for plant organ growth and cell enlargement research.

**Supplementary Materials:** The following supporting information can be downloaded at: https://www.mdpi.com/article/10.3390/horticulturae8100865/s1, Table S1. Different expression genes of the pedicel AZs at flower stage (F) and 5 DPA in tomatoes. Table S2. Different expression genes of the pedicel AZs at flower stage (F) and 15 DPA in tomatoes. Table S3. Different expression genes of the pedicel AZs at flower stage (F) and 30 DPA in tomatoes. Table S4. Different expression genes of the pedicel AZs at 5 DPA and 15 DPA in tomatoes. Table S5. Different expression genes of the pedicel AZs at 5 DPA and 30 DPA in tomatoes. Table S6. Different expression genes of the pedicel AZs at 15 DPA and 30 DPA in tomatoes. Table S7. The primers for qRT-PCR. Table S8. The raw Ct values of qRT-PCR. Table S9. The information of the DEGs that were annotated in each of the GO terms. Figure S1. The enrichment analysis for genes involved in the plant hormone pathways during pedicel AZ development in tomatoes. Figure S2. The enrichment analysis of the DEGs belonging to different classes of auxin response genes.

**Author Contributions:** H.W. conceived and designed the experiments. X.D. and Y.W. conducted the experiments, analyzed the data (with assistance from Y.Y.), and wrote the manuscript. H.W. revised the manuscript. All authors have read and agreed to the published version of the manuscript.

**Funding:** This study was supported by the "Pioneer" and "Leading Goose" R & D Program of Zhejiang (no. 2022C02051); the Zhejiang Natural Science Foundation of China (grant no. LY19C150008); the Opening Project Fund of the Key Laboratory of Biology and Genetic Resources of Rubber Tree, Ministry of Agriculture and Rural Affairs, PR China/State Key Laboratory Breeding Base of Cultivation and Physiology for Tropical Crops/Danzhou Investigation and Experiment Station of Tropical Crops, Ministry of Agriculture and Rural Affairs, PR China (no. RRI- KLOF202102); the Natural Science Foundation of Zhejiang province (grant nos. LY21C150002 and LQY19C150001); the National Natural Science Foundation of China (grant nos. 31872105, 31972221, 32002048, 31801862, and 32172595); the National College Students Innovation and Entrepreneurship Training Program in 2019 and 2021 (nos. 202110341043 and 201910341005); the Student Scientific research training program of Zhejiang Agriculture and Forestry University (nos. 2021KX0196 and 2021KX019), Ministry of Agriculture; and the National Key Research and Development Program of China (nos. 2018YFD1000800 and 2019YFD1000300).

**Institutional Review Board Statement:** Not applicable.

**Informed Consent Statement:** Not applicable.

**Data Availability Statement:** The data that support the findings of this study are openly available in the National Center for Biotechnology Information (NCBI) SRA database under the BioProject ID: PRJNA875968.

**Conflicts of Interest:** The authors declare no conflict of interest.

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
