# Peer review of "Transcriptome Analysis of the Development of Pedicel Abscission Zone in Tomato"

_horticulturae, doi:10.3390/horticulturae8100865_

Round 1

Reviewer 1 Report

The manuscript by Dong and colleagues presents comprehensive transcriptomic approach to understand the molecular mechanisms of pedicel abscission zone in tomato plants. The authors performed Illumina-based RNA sequencing of tomato pedicels at different DPA stages. The analysis identified several significant changes in transcript levels during 0 to 5 DPA, 5 to 15 DPA, and 0 to 15 DPA. Subsequent bioinformatic analyses characterized them by GO and KEGG enrichment analyses. The authors also validated the expression patterns of auxin-related genes, like AUX/IAA, by qRT-PCR. As a whole, the study seems to be technically sounds. The topic is timely and interesting. However, I had some concerns about the method description and the reproducibility as follows.

Methods:

1) Illumina-based RNA sequencing

There was no information about the used platform, the quality control, filtering low-quality read data, and read mapping. This is not reproducible for readers.

2) Tomato genome

Please add the details of tomato genome sequences used in this study. For example, assembly (annotation) version?

3) Literature citation

Please cite the latest paper regarding software and databases used in this study. For example, DESeq, GO, and KEGG.

4) There was no description regarding PCA and GO/KEGG enrichment analysis (see Fig. 2). For example, did you use data transformation and data scaling methods in PCA? Regarding enrichment analysis, did you use p-value correction for multiple testing problem?

5) Would you please open your transcriptome data in community-approved data repository like NCBI SRA? Please describe the accession number in the section “Data Availability Statement.”

6) Fig. 7 Please describe the plots. For example, what are error bars? Mean or average/median?

Author Response

The attachment below is our response to reviwer 1

Reviewer 2 Report

Below are my comments and suggestions in details.

Intro:

1)    A number of genes have been reported to be involved in the differentiation and development of AZs (reference is required).

2)    Please provide more details on the role of other hormone-related genes in addition to Auxin genes. 

M&M:

1)    please describe details of the biological replicates. Are these pools of different plants? Are they taken by the same plant? Is a single replicate corresponding to a plant or a pool of them?

2)    Please deposit RNA-seq data in public repository. 

3)    Chapter 2.8 qRT-PCR: Relative expression of qRT- PCR was calculated by each experiment was performed independently two times with at least three biological samples. How data are analyzed? Could you provide raw Ct values?

Results:

1)    Relative expression of qRT- PCR was calculated by each experiment was performed independently two times with at least three biological samples. This depends on the type of replicates that has been considered. If replicates are within the same plant the variability should be reduced, please explain details in relation to previous comments on M&M.

2)    interestingly, seven pathways—“photosynthesis”, “alpha-linolenic ”, “glycosphingolipid biosynthesis-lacto and neolacto series”, “glyco- sphingolipid biosynthesis-globo and isoglobo series”, “glyoxylate and dicarboxylate me- tabolism”, “brassinosteroid biosynthesis”, and “carotenoid biosynthesis”—were not sig- nificantly changed during both 0–5 DPA and 5–15 DPA, whereas they were significantly altered during 0–15 DPA; and two pathways—“amino sugar and nucleotide sugar metab- olism” and “plant-pathogen interaction”—were changed during both 0–5 DPA and 5–15 DPA, but not in 0–15 DPA. Please describe in Discussion why these different and not apparently agreeing findings were reported. Make a hypothesis about it. 

3)    Table 1 and 2: Instead of absolute Fpkm, it will be good show the relative values of the different pairwise comparisons to confirm that these genes are above the defined threshold. Explain better what you mean with absolute values of fold change and why you choose 5 as a threshold. 

4)    3.5. GO and KEGG Enrichment Analysis of the Key DEGs under Three Comparisons – Are these data from the commonly DEGs between different paiwise comparisons? All three stages? Please explain better. In addition, the GO-terms are very general (on the top of the GO tree), could you focus on the indication of most specific GO-term (downstream). 

5)    Explain why KEGG and GO enrichment analysis did not show GO-term related to auxin pathways. 

6)    In the Discussion it is specified that SAUR is the most associated with AZ development based on the fact that the number of genes is the highest? Could this do the fact that this family is the biggest in the tomato genome? Please perform an enrichment analysis or any statistical method that may associate a significant difference of expression among the different classes. Also please explain why different genes of the same category show a different pattern of expression.  

7)    I suggest to perform a protein-protein interaction analysis (es. Using STRING) to find out which proteins interact with auxins (querying the analysis with the list of regulated auxin genes). Also, it would be good to find out if these proteins interacting with auxin genes are in the list of DEGs of the different pairwise comparisons. Discuss these additional findings. 

8) Please perform a gene set enrichment analysis  for genes involved in all hormone pathways in order to find out the crosstalk between auxin and the other home categories. 

Author Response

The attachment below is our response to reviwer 2.

Round 2

Reviewer 1 Report

The manuscript has much improved, and the authors did a good job to address the constructive suggestions of the reviewer.

Reviewer 2 Report

The authors have addressed my comments.